# Solvent Swelling-Induced Halogenation of Butyl Rubber Using Polychlorinated N-Alkanes: Structure and Properties

**DOI:** 10.3390/polym15204137

**Published:** 2023-10-18

**Authors:** Ksenia Valeriyevna Sukhareva, Nikita Romanovich Sukharev, Irina Ivanovna Levina, Peter Ogbuna Offor, Anatoly Anatolyevich Popov

**Affiliations:** 1Higher School of Engineering, Plekhanov Russian University of Economics, 36 Stremyanny Ln, 117997 Moscow, Russia; popov.ana@rea.ru; 2Institute of Biochemical Physics Named after N.M. Emanuel, Russian Academy of Sciences, 4 Kosygin St., 119991 Moscow, Russia; niiikos007@gmail.com (N.R.S.); iilevina@inbox.ru (I.I.L.); 3Metallurgical and Materials Engineering Department, University of Nigeria, Nsukka 410001, Nigeria

**Keywords:** halogenation, butyl rubber, polychlorinated n-alkanes, mechanochemical modification, thermal and oxidative stability, chemical resistance, mechanical properties

## Abstract

Traditional butyl rubber halogenation technology involves the halogenation of IIR using molecular chlorine or bromine in a solution. However, this method is technologically complex. This study investigated a novel method for the halogenation of butyl rubber to enhance its stability and resistance to thermal oxidation and aggressive media. The butyl rubber was modified through mechanochemical modification, induced by solvent swelling in a polychlorinated n-alkane solution. During the modification, samples were obtained with chlorine content ranging from 3 to 15%. After extraction, the halogen content was quantitatively determined with the oxygen flask combustion method and X-ray photoelectron spectroscopy. It was shown that for samples with total chlorine content of up to 6%, there was almost no leaching of chlorine from the samples. The chemical structure of the extracted rubbers was ascertained using FT-IR and ^1^H NMR spectroscopy, and it was demonstrated that all samples showed absorption peaks and signals typical for chlorobutyl rubbers. It was observed that modification with polychlorinated n-alkanes improved the thermal and oxidative stability (the oxygen absorption rate decreased by 40%) and chemical resistance, estimated by the degree of swelling, which decreased with the increase in the chlorine content. This technology allows the production of a chlorinated rubber solution that can be directly used by rubber goods manufacturers and suppliers.

## 1. Introduction

The chemical modification of elastomers plays a crucial role in enhancing the physical properties of rubbers [1]. Today, the demand for elastomeric materials with specific properties, ensuring performance under extreme conditions, is continually increasing. Owing to their unique mechanical properties, such as high deformability across several temperatures and energy absorption capabilities, elastomers constitute a significant material class applicable in various fields. They find utility in a broad spectrum of applications, ranging from footwear to space vehicles. In this context, the necessity to develop innovative production technologies for diverse industrial sectors is intensifying. The bulk modification of rubber offers significant technological advantages. Typically, this involves the incorporation of diverse additives to the unvulcanized rubber or adjustments made during the latex stage. The additives can include a range of substances, including inorganic fillers (e.g., silica [2,3], calcium carbonate), maleic anhydride [4,5], and zirconium phosphate [6]. Additionally, materials like organoclays [7], graphene particles [8], bentonite nanoparticles [9], polyolefins [10], and carbon nanotubes [11] can also serve as components for modified rubber. Among the several types of rubber modification, halogenation reactions are one of the most common and effective, imparting versatile curing properties that enhance joint vulcanization, adhesion, and resistance to aging and different chemicals. Regarding the study of a novel chlorination method for butyl rubber, several aspects related to the development of various technological solutions in the field of halogen modification of elastomers are analyzed.

Butyl rubber (IIR) is a synthetic rubber, a copolymer of isobutylene with a small amount of isoprene. On the basis of the high saturation bond and closely stacked methyl groups of the polymer chain, butyl rubber exhibits outstanding air retention and heat and ozone resistance compared with various rubbers, including natural and styrene–butadiene rubbers. Halobutyl rubbers were first investigated in the 1940s, and, in the 1950s, they were initially produced on an industrial scale. The most important commercial butyl derivatives are bromobutyl (BIIR) and chlorobutyl (CIIR). Halobutyl rubbers (CIIR and BIIR) offer valuable performance characteristics like IIR, such as low gas permeability and ozone stability, but differ in terms of their increased resistance to atmospheric action, UV light, ozone, and many chemical reagents, as well as enhanced dynamic properties. The addition of halogen to IIR permits greater curing versatility, providing diverse choices in cure systems to enhance covulcanization, adhesion, and the degree of cross-linking. The most common commercially available halobutyl rubbers nowadays include CIIR 1068 and 1066 (EXXON), CIIR 1240 (Lanxess), and CIIR HBK-139 (Niznekamskneftehim), with a chlorination concentration of 1.1–1.4 wt.%, and BIIR BB2030 (Lanxess), with a bromination concentration of 1.8–2.5 wt.%. Halobutyl rubbers with higher halogen content are not currently industrially produced. Currently, the only method used in the industry to produce halobutyl rubbers involves the halogenation of IIR using molecular chlorine or bromine in a solution. In the production process of halobutyl elastomers, the halogen is introduced into a butyl solution in hexanes, operating at temperatures ranging from 40 °C to 65 °C. The actual mechanism of substitution is believed to involve a cationic cyclic intermediate, within which the halogen becomes bonded to the isoprenyl double bond. Subsequently, various microstructures are generated, the distribution of which depends on the specific halogen type. The primary structure consists of exomethylene (EXO) units (comprising 70–90%), with smaller quantities of endomethylenes (constituting 10–20%). The neutralization of the resulting acid is accomplished by adding a diluted aqueous caustic solution. This solution is then separated and extracted before proceeding with subsequent stages of processing. Stabilization plays a critical role in halobutyl rubbers, preventing premature cross-linking and thermal dehydrohalogenation. CIIR maintains many of the desirable characteristics of IIR, such as low gas permeability as well as good thermal and oxidative stability. CIIR and BIIR offer excellent resistance to weathering, ozone, and hot air; exceptionally good resistance to acidic and basic chemicals; extremely low permeability to gases and liquids; and commendable rheological properties. BIIR is used to produce tire inner liners and sidewalls and is used in tire treads. It is also suitable for use in pharmaceutical stoppers and rubber articles requiring good resistance to chemicals, weathering, and ozone, such as tank linings, conveyor belts, and protective clothing.

The analysis of existing production facilities showed that the industrial methods of obtaining halogenated IIR are technologically complex. These processes include several stages that are power-consuming:Chlorination or bromination of the IIR solution with gaseous halogen;Neutralization of the formed halogenated IIR;Washing of the halogenated IIR solution to remove salts;Introduction of a stabilizing antioxidant into the halogenated IIR;Degassing, isolating the halogenated IIR, and drying.

The main disadvantages of traditional butyl rubber halogenation technology include the use of a specific butyl rubber grade with high Mooney viscosity and unsaturation values for the production of CIIR. Additionally, there is the need to recover bromine and chlorine from water streams during the stages of excess halogen neutralization and the washing of the BIIR solution. The optimization of the current production technology for halogen-containing butyl rubbers aims to streamline the process stages, enhance profitability, improve environmental sustainability, and investigate alternative halogenation systems. Abramova et al. reported a study on direct and oxidative IIR halogenation using systems based on tert-butyl hypochlorite (TBHC) [12]. However, as TBHC is an unstable product, its transportation presents challenges due to its potential for decomposition when exposed to heat or radiation. Maksimov et al. described a method combining the electrochemical production of halogen with the halogenation of rubber within the reaction chamber of the electrolyzer [13]. Meanwhile, Orlov et al. halogenated IIR using aqueous solutions of halogens produced electrochemically [14]. The efficiency of the bromination reaction of butyl rubber can be increased by using NaClO in a mixture with Br_2_, in which NaClO acts as an oxidant and can react with HBr (the byproduct in the bromination reaction) to oxidize it back to molecular bromine. The regenerated bromine can continue to react with the carbon–carbon double bond [15]. In the work [16], the influence of acidic and basic environments on the formation of bromide structures during the synthesis of bromobutyl rubbers was demonstrated.

The method of butyl rubber modification discussed in this study is based on a mechanochemical approach to initiate the modification process. The method of halogenation of butyl rubber examined in this work represents the development of an approach to mechanochemical solid-phase halogenation, laid out in the works of Andriasyan et al. [17,18]. The principle was based on solid-phase mechanochemical halide modification, initiated by shear stresses during rubber processing in a rubber mixer in the presence of chlorinated paraffins. The method of mechanochemical solid-phase halide modification, initiated by shear stresses, has significantly improved the technology of halogenating elastomers by reducing the number of stages, speeding up the process, and increasing the level of technological and environmental safety of production. However, it has the following disadvantages:Efficient mixing of the viscous elastomer mass requires high energy consumption;There is difficulty in obtaining a homogeneous product;The process is accompanied by the self-heating of the reaction mass; therefore, strict process control is required, and, if necessary, measures to cool the mixing equipment;This method is limited in terms of the halogen content in the modified rubbers.

Based on the accumulated literature data on the consideration of rubber swelling as a mechanochemical process during which the polymer experiences high stress, the possibility of conducting mechanochemical halide modification under the influence of swelling pressure was studied. Recent publications also indicate that the swelling of polymer networks can generate tension forces potent enough to drive bond cleavage reactions [19,20,21,22]. The literature regards the concept of an entangled network of elastomers as a mesh of entangled chains [23,24]. Previously, it was shown that during the swelling process, additional osmotic forces act on the polymer chains, which can induce their swelling-driven mechanochemical activation. These forces lead to the macroradical rupture of the entangled network in the rubber materials. The swelling of uncross-linked rubber depends on the polymer’s nature, the number of trapped entanglements (referring to the presence of chain junctions, steric restrictions, and entanglements), the phase state, and the distribution of crystalline formations in its volume. Gravimetric analysis and fluorescence imaging demonstrated that, for most solvents with intermediate polarity, swelling and activation are directly linked, showing that the forces acting on the mechanophore due to swelling result in the mechanical activation of the spiropyran mechanophore [21]. The spiropyran mechanophore, serving as a force-sensitive molecule, was used to study the generation of mechanical forces at the molecular level during the solvent swelling of polymers, specifically cross-linked polymethyl methacrylate (PMMA). These mechanical forces lead to solvent swelling-induced covalent bond breakage [25]. The initiation of covalent bond scission likely comes from high stresses at the diffusion front, anticipated to be approximately 10 MPa. This observation aligns with the chain scission mechanism [26] and reduced viscosity due to swelling [27]. In a study [28], multiquantum NMR spectroscopy revealed signals from uncross-linked components of the polymer chain, which were formed by topological constraints and entanglements that represented topological confinement in the motion of one polymer chain by other chains. Such a conclusion has received empirical confirmation across several polymer systems and through computer simulations [29,30]. The heterogeneity of the solvent transfer process to the rubber structure results in absorption rate gradients, which in turn lead to stresses in the macrochains at the boundaries of these microregions. Even beyond these regions, in the presence of the entangled network, overstresses appear in stretched areas during swelling when the solvent absorption rate surpasses the time needed to untangle the macromolecular loops and entanglements. This rubber swelling leads to the development of mechanical stresses in the macromolecular chains and the eventual mechanodestruction and mechanoactivation of the rubber chains.

Halogenoalkanes like CCl_4_ can react with radical substances and serve as reagents in free radical chain and topomerization reactions [31]. During swelling, the mechanoactivated macromolecules of rubber can interact with halogenoalkanes. This can result in halogen atoms and groups being incorporated into the polymer chains through swelling induction. In this study, such an effect is used for the halogenation of IIR. Moreover, the technology holds practical significance because chlorinated butyl rubbers are often required in solution form for applications such as adhesives and protective rubber coatings. This technology allows one to produce a chlorinated rubber solution that can be directly used for rubber goods manufacturers and suppliers to produce adhesive bonds, rubber coatings, or for spreading on fabrics when creating rubber pneumatic structures. The advantage is that one can obtain chlorinated butyl rubber based on readily available butyl rubber and an accessible chlorine-containing modifier, replacing industrial chlorinated butyl rubber (CIIR) in production. Thus, rubber goods manufacturers can partially replace industrial CIIR with modified butyl rubber in products where a solution of chlorinated butyl rubber is required, without compromising the quality of the product.

## 2. Materials

IIR (IIR-1675n grade, PJSC Nizhnekamskneftekhim, Russia), a copolymer of isobutylene and isoprene, was used as a basic raw material.

CIIR (CIIR-139 grade, PJSC Nizhnekamskneftekhim, Russia) was obtained by the chlorination of butyl rubber with minimum unsaturation of 1.8 mol%. The structure of the butyl rubber, exo-methylene halobutyl rubber, and endo-halomethyl butyl rubber are displayed in Figure 1. The properties of IIR and CIIR are displayed in Table 1.

Chlorinated paraffin (CP) is a complex mixture of polychlorinated n-alkanes that differ in chain length composition and degree of chlorination. Technical CP mixtures (CP-66T) (AO Kaustik, Volgograd, Russia) have chlorine content of 69.5 ± 0.1 wt.% and are yellow-tinged and partly clotted powders. CP is produced by the chlorination of alkane mixtures, which forms complex products of thousands of homologs and congeners. The CP-66T product has, on average, one chlorine substituent on almost every carbon atom. According to the alkane chain length, CP-66T is classified as long-chain CP. Chlorinated paraffins are widely available raw materials classified as low-hazard substances and do not require additional protective measures during handling (unlike gaseous chlorinating compounds). Previously, the effectiveness of using this CP for halogenation purposes was demonstrated [32,33].

## 3. Methods

### 3.1. Modification in Polychlorinated N-Alkane (CP) Solution

The modification of IIR was performed by dissolving CP in toluene at different CP ratios, as listed in Table 2. The mean carbon formula of CP indicates the approximate mean number of chlorine and hydrogen atoms per carbon atom. On the basis of the mean carbon formula, 1 g of CP contains 0.695 g Cl. The theoretical calculation of the amount of introduced chlorine atoms was based on data regarding the chlorine content in chlorinated paraffin. After the complete dissolution of the CP in toluene, IIR (10 g) was cut into 0.5-cm^2^ pieces and placed in a flask. The IIR pieces were kept in a solution of toluene (70 mL) and CP. The dissolution of rubber was performed while mechanically stirring it using a magnetic stirrer at 400–500 rpm for 3 days to obtain a uniform glue solution. The solution mixture was cast into a glass Petri dish. Finally, the solvent was evaporated at room temperature until a constant weight was obtained. Chlorination was performed at a temperature of 22 ± 2 °C.

### 3.2. Extraction

To determine the bound chlorine in the composition of the rubber macromolecules, the samples of the chlorinated rubber were subjected to extraction to remove the unreacted modifier. Extraction was performed in a Soxhlet apparatus by adding acetone for 20 h [34]. Acetone was chosen as the solvent because the modifier was highly soluble in it and the rubber itself did not dissolve in it. The apparatus for Soxhlet extraction, including a condenser, tubes, the Soxhlet chamber, a round-bottom flask, and a heating mantle, was set up for solvent extraction.

### 3.3. Oxygen Flask Combustion Method

The oxygen flask combustion method is based on burning rubber in a flask filled with oxygen, followed by titration with a solution of mercury (II) nitrate in the presence of diphenylcarbazone [35]. This method is one of the most used methods in determining the total amount of chlorine contained in a CIIR. This test is based on the use of the Thomas–Schoniger combustion flask. For this method, a weighted sample of CIIR is wrapped in a piece of ash-free filter, and this wrapped sample is placed inside a platinum sample carrier. A flask of 500 mL is filled with pure oxygen by flowing this gas for 3–5 min. A solution to absorb chlorine is set inside the flask. Then, the filter is burned with a flame and immediately placed inside the flask.

The total chlorine content (wt.%) was calculated using the following equation:$$\left[Cl\right]=\frac{V_{1}-V_{2}}{m}\times K_{c}\times 100,$$
where *V*_1_ and *V*_2_ are the volumes of 0.02 N Hg(NO_3_)_2_ solution used for the titration of the working and control solutions in milliliters, and *m* is the sample weight.

The correction coefficient *K_c_* used to adjust the concentration of the Hg(NO_3_)_2_ solution precisely to 0.02 N was calculated using the formula
$$K_{c}=\frac{m}{1.1690\times V}$$
where

*m*—mass of NaCl, mg;

*V*—volume of the 0.02 N solution of Hg(NO_3_)_2_ used for titration, mL;

1.1690—mass of NaCl corresponding to 1 mL of 0.02 N solution of Hg(NO_3_)_2_, mg.

### 3.4. X-ray Photoelectron Spectroscopy

X-ray photoelectron spectroscopy (XPS) analysis was performed on a JSM-U3 scanning electron microscope (JEOL, Tokyo, Japan) at an accelerating voltage of 8 keV, equipped with an energy-dispersive spectrometer, EUMEX (Hamburg, Germany).

### 3.5. ^1^H NMR Spectroscopy

^1^H NMR spectra were acquired in CDCl_3_ (^1^H 7.26 ppm) at 500 MHz on a Bruker Avance-500 spectrometer. All chemical shifts were referenced to residual chloroform in deuterated chloroform (CDCl_3_). The concentration of the solutions was maintained at 5 wt.%. NMR data were processed using the Mestrenova software (V14.2.1-27684).

### 3.6. FT-IR Spectroscop

Fourier transform infrared spectroscopy (FT-IR) was used to collect infrared spectra of butyl rubber before and after modification. The spectra were recorded on the Bruker Lumos IR Fourier microscope (Bruker Corp., Bremen, Germany) at the temperature of 24 ± 2 °C in the range of wave numbers 4000 ≥ ν ≥ 500 cm^−1^. FT-IR data were processed using the SpectraGryph 1.2 software.

### 3.7. Preparation of CIIR (IIR) Vulcanizates

The IIR, CIIR (commercial grade), and CIIR-3, CIIR-6, CIIR-9, and CIIR-15 vulcanizates were prepared on the basis of the compounding formulations shown in Table 3. The quinol ester of p-quinone dioxime (trade name EH-1, «Minkar», Russia) is a powder with a light-yellow to dark-yellow color [36]. The dithiophosphate accelerator (trade name Kvalaks C1, “NPP Kvalitet”, Moscow, Russia) is a transparent liquid ranging from yellow to brown in color, with a zinc mass fraction of 8.8% and a kinematic viscosity at 100 °C of not less than 6 cSt. Previously, it was established that the p-quinone dioxime (6 phr) and the dithiophosphate accelerator (2 phr) correspond to the composition of the optimal vulcanizing group for the solution of chlorinated butyl rubber [37].

The rubber was mixed on a two-roll mill. Subsequently, the accelerator (dithiophosphate accelerator) and curing agent (quinol ester of p-quinone dioxime) were added to the rubber compound. The resultant compounds were vulcanized using a hydraulic press operated at 4 MPa pressure at the most appropriate cure temperature (150 ± 5 °C) and time (5 min).

### 3.8. Oxidative Degradation

A manometric method was used for the kinetic study of rubber oxidation. The oxidation rate was determined by the amount of oxygen captured using a manometric device. Rubber oxidation was conducted in the temperature ranges 150 ± 2 °C and 180 ± 2 °C, and p(O_2_) = 300 torr. Solid KOH was used to absorb volatile oxidation products. The experiment was carried out for 330 min. The duration of samples’ testing using the manometric oxidation method depends on the objects under study and is not strictly regulated. The time is determined based on how easily the samples oxidize (consume oxygen). In this case, the experiment concluded at 330 min, as further continuation of the experiment could have led to a decrease in the method’s sensitivity due to oxygen depletion in the reaction cells (oxygen is supplied to the system only at the beginning of the experiment). As oxygen is consumed, the concentration and pressure of oxygen deviate from the initial values, and the process proceeds at lower oxygen concentrations in the reaction cell compared to the beginning of the experiment.

### 3.9. Mechanical Properties

Determination of the tensile strength properties of the samples was performed using a Devotrans DVT GP UG 5 universal testing machine (Turkey) in accordance with the ISO 37:2017 standard [38], at a testing rate of 100 mm/min. Samples were cut out on a pneumatic punching press GT-7016-AR (GOTECH testing Machines Inc., Istanbul, Turkey) Each data point was corroborated with five measurements. Resilience was measured using a Schob pendulum, following the ASTM D7121 testing methodology. Furthermore, hardness was ascertained using a Shore A type durometer, adhering to the testing method outlined in the ASTM D 2240-05 (2010) standard [39].

### 3.10. Chemical Resistance

The chemical resistance of rubbers was estimated by the swelling test change in mass (after immersion) according to ASTM D471. As immersion media, we used mineral-based engine oil, diesel fuel, nitric acid (HNO_3_), and toluene. The samples were weighed every 24 h. The swelling degree (%) was calculated by the following equation:Swelling degree = (W_2_ − W_1_)/W_1_ × 100 [%],
where W_1_ is the weight of the polymer before swelling, and W_2_ is the weight of the polymer after swelling.

## 4. Results and Discussion

### 4.1. Microstructural Analysis of the IIR with Different Amounts of CP

Samples of butyl rubber with added CP (CIIR-3, CIIR-6, CIIR-9, and CIIR-15) were Soxhlet-extracted in a Soxhlet apparatus for 20 h to remove the unreacted CP, which was chemically unbound and, consequently, easily washable with acetone from the film of the modified rubber. The presence of chlorine in the modified rubber samples was confirmed using two parallel methods: mineralization of the rubber and chemical analysis of the collected chlorine (Table 4) and X-ray photoelectron spectroscopy (Table 5).

The results of the analysis using the mineralization of CIIR-3, CIIR-6, CIIR-9, and CIIR-15 and chemical analysis of the collected chlorine (Table 4) and X-ray photoelectron spectroscopy (Table 5) showed remarkably similar results. A chemical analysis of the collected chlorine indicated that samples with halogen content of up to 6 wt.% inclusive did not have unreacted CP. A further increase in the amount of introduced CP of ≥9 wt.% of chlorine led to the preservation of a small amount of unreacted modifier (CP). X-ray photoelectron spectroscopy revealed that in all samples there was approximately the same leaching of the unreacted halogen from the sample at approximately 40–50%.

### 4.2. ^1^H NMR Spectroscopy

NMR spectroscopy was used to quantitatively study the microstructure of IIR, CIIR-3, CIIR-6, CIIR-9, CIIR-15, and CIIR (commercial grade) and the bonding structure of chlorine with the molecular chain. The ^1^H NMR spectra are shown in Figure 2, Figure 3, Figure 4 and Figure 5.

The aliphatic region (0.9–1.95 ppm) of the sample spectra was considered, wherein the resonance of protons from methylene and methyl groups of the isoprene and isobutyl IIR units was observed [40]. According to Figure 2, the aliphatic CH_3_ and CH_2_ signals of the isobutylene structural units appeared at δ 0.98 (weak signal) and δ 1.09 and δ 1.40 ppm (intense signals), respectively [41]. In the ^1^H NMR spectra of IIR, two weak single peaks at δ 1.63 and δ 1.92 in the aliphatic region were found, corresponding to the aliphatic CH_3_ and CH_2_ signals of isoprene units, respectively [42].

Proton signals from halogenated units CIIR (commercial grade), CIIR-3, CIIR-6, CIIR-9, and CIIR-15 are in the 4.0–5.8-ppm region and do not overlap with signals from the isobutylene units. Single peaks at 5.36 and 5.01 ppm stem from protons of the carbon–carbon double bond of the exo-methylene unit (EXO). The single peak at 4.20 ppm corresponds to a distinct triplet from the -CHCl- group. This signal barely overlaps with other signals, designating it as the unique signal for the EXO form. The chemical shift range 4.97–5.09 includes a triplet and a singlet at 5.07 and 5.01, respectively. In all chlorinated IIR, a triplet at 5.07 was overshadowed by a strong singlet at 5.01 ppm, so this signal could not be integrated separately with sufficient accuracy [15,43]. Isoprenyl units of butyl rubber predominantly exist in a single isomeric form. This is evident from the presence of only one signal in the 1.65 and 1.94 ppm regions of the spectrum displayed in Figure 2. A single strong signal at 5.07 ppm indicates the presence of hydrogen at the allylic carbon atom and corresponds to the signal from isoprene fragments of butyl rubber (Figure 3). The single triplet at 5.05 ppm is almost invisible in CIIR samples; the signal is barely noticeable in the CIIR-6 sample and overlaps with the signal at 5.01 ppm in the CIIR-9 and CIIR-15 samples. The signal at 5.05 ppm determines the residual unsaturation of the samples, which is significantly lower in CIIR samples compared to the neat IIR. The presence of either 1,2- or 3,4-enchained isoprenyl units in significant quantities would necessitate additional signals in this spectrum region. Because of halogenation, the bound isoprene units transform into their corresponding halo-substituted forms.

Upon adding a baseline (red line) to the CIIR-9 and CIIR-15 spectra, the 2.4–3.5 and 4.27–4.97 ppm regions overlapped with unresolved clusters of low-intensity multiplets. It seems that these peaks pertain to the residual amount of the chlorine modifier, which persisted in the samples mixed physically with it. For reference, the spectrum of CP was used (Figure 4). The ^1^H NMR spectra of the chlorine modifier revealed two distinct unresolved clusters of multiplets: the “unchlorinated” range (δ(^1^H) = 0.8–3.2 ppm) and the “monochlorinated” range (δ (^1^H) = 3.2–5.3 ppm). The relative area of the “monochlorinated” range in ^1^H NMR = 1.07, and the Cl content = 71.7 ± 0.1. Thus, the emergence of an additional signal cluster might be linked to the presence of the unreacted and unextracted chlorine modifier in the rubbers.

A comparison of the ^1^H NMR spectra of all modified rubbers and CP revealed that additional signals from the residual modifier were observed only in CIIR-9 and CIIR-15 (Figure 4 and Figure 5 and Table 6).

This is consistent with the chemical analysis data obtained using the oxygen flask combustion method. The halogenation of butyl rubber, particularly at the dominant trans-1,4-incorporated isoprene units, results in various enchained microstructures. The most prevalent structure is the EXO allylic halide. A secondary endo-methylene (ENDO) allylic halide, either in the trans or cis configuration, is also produced but at lower concentrations [44]. As many signals in the ^1^H NMR spectra partially overlap with the residual signals from CP, a quantitative assessment of the chlorine concentration component seems unfeasible. The total area under the peaks from 1.5 to 0.67 ppm (isobutylene area free from the residual signals of CP) was integrated and normalized to a count of 100. Peaks in the δ 4.23 to δ 4.16, δ 5.05 to δ 4.98, δ 5.4 to δ 5.29, and δ 5.5 to δ 5.46 ppm regions were integrated. The normalized ^1^H NMR integration of these peaks was used to provide the relative concentrations of the halogenated units (Table 7).

### 4.3. FT-IR Spectroscopy

The FT-IR spectra of the as-received and modified rubber were recorded on a Fourier transform infrared spectrophotometer in the region of 4000–500 cm^−1^ using a thin film of the modified polymer. The FT-IR spectra of butyl rubber and chlorinated butyl rubber are shown in Figure 6.

The analysis of this spectrum reveals the following characteristic absorption bands: as-received IIR showed characteristic symmetric and asymmetric stretching vibrations attributed to CH_2_ and CH_3_ (CH_3_, 2951 cm^−1^; CH_2_, 2875 cm^−1^; CH_3_, 2894 cm^−1^). For analysis, the region of 450–800 cm^−1^ was taken (Figure 6). C–Cl absorbances are commonly found within the range of 600–800 cm^−1^. Three IR absorption bands at 695 and 530 cm^−1^ clearly reveal the presence of the C–Cl bond [45]. The peak corresponding to the C–Cl bond in a wave number range of 690–695 cm^−1^ was observed in all CIIR (CIIR-3, CIIR-6, CIIR-9, and CIIR-15) samples [46].

### 4.4. Thermal and Oxidative Stability

The oxidation of a polymer is typically accompanied by oxygen absorption. Therefore, measuring the amount of oxygen absorbed by the polymer is a primary method of studying the oxidation process. Oxygen absorption by the samples was gauged at specific time intervals, and the results were used to plot kinetic curves. The rate of thermal oxidation of polymers is typically higher than that of their thermal decomposition. Thermal oxidation often leads to detrimental changes in polymer properties, making it a crucial factor in determining the service life of many rubber materials. The study of thermal oxidation was conducted on unfilled and uncross-linked systems. The thermal and oxidative behavior and the corresponding kinetic curves of oxygen absorption for cured IIR and CIIR samples are depicted in Figure 7, Figure 8 and Figure 9.

An analysis of the isothermal oxidation of rubber in the temperature range of 150–180 °C confirms the following main regularities of this process.

Oxidation proceeds without an induction period (Figure 7 and Figure 8); IIR and CIIR have no initiation time and begin to oxidize immediately after the start of the experiment.The process rate (Figure 9) is quite different: for as-received IIR, it is 0.61 × 10 ^−5^ and 2.6 × 10^−5^ mol/kg·s for 150 °C and 180 °C, respectively; for chlorinated IIR, it ranges from 0.46 to 0.53 × 10^−5^ mol/kg·s and from 1.55 to 1.75 × 10^−5^ mol/kg·s for 150 °C and 180 °C, respectively, depending on the chlorine content.Oxidation proceeds at sufficiently high rates.

In the case of CIIR, the halogenated structures display more oxidative resistance than the isoprene units in IIR. Moreover, an increase in the chlorine content of >9 wt.% does not lead to an increase in oxidative stability. The oxidation rate of CIIR-15 at a temperature of 180 °C was higher because of the intensified dehydrochlorination process accompanied by the elimination of HCl. The CIIR samples with higher chlorine content are more likely to release HCl at the experimental temperature. Chlorine can generate additional stable cross-links between the polymer chains and has a key role in increasing the thermal stability of the elastomeric materials. It is observed that the amount of oxygen absorbed by the chlorinated samples decreased as the period of exposure in the oxygen atmosphere rose.

### 4.5. Chemical Resistance

Among the distinct types of chemical modifications in rubbers, halogenation reactions are the most common and effective ones, enhancing the solvent and oil resistance. Figure 10 presents the variation in the percentage swelling of the as-received IIR and chlorinated IIR (CIIR-3, CIIR-6, CIIR-9, and CIIR-15) after immersion in a petroleum solvent, nitric acid, toluene, and engine mineral oil for varying durations (24, 72, and 144 h) in the temperature range 22 ± 2 °C. The swelling in rubber specimens is described in terms of the degree of swelling, which is defined by the ratio of the volume of the swollen specimen to that of a dry specimen. Changes in the properties of rubber during swelling are attributed to the diffusion of liquid molecules into the intermolecular spaces of the rubber and the weakening of its intermolecular bonds.

According to the results illustrated in Figure 10, the aggressive media resistance of halogenated CIIR is significantly higher than that of the neat IIR. The degree of swelling decreases with the increase in the chlorine content. The oil resistance as reflected by the percentages of mass change in oil immersion showed that rubber sheets prepared from CIIR exhibited better oil resistance relative to the control as-received IIR. Comparatively, the petroleum resistance between CIIR-3 and CIIR-6 was nearly similar. The swelling resistance of rubber depends on the extent of the elastomer’s polarity. The more polar groups attached to the rubber molecules, the higher the polarity and the better the swelling resistance. The decline in petroleum solvent, oil, aromatic solvents, and acid absorption can be explained on the basis of the increased polarity of CIIR. The inclusion of halogen groups introduces a dipole moment in the elastomer chain. Furthermore, chlorine atoms serve as cross-linking junctions that help to establish the chain network. The interaction of these centers with ZnO could be the reason that a denser and stronger chain network is formed during the curing process [47]. This results in the restricted diffusion of oil, gasoline, and acid molecules into the halogenated rubber.

### 4.6. Mechanical Properties

The data from all mechanical tests were processed and evaluated. The influence of chlorine content on the mechanical properties of rubber blends is presented in Table 8.

The mechanical characteristics of these mixtures are inferior to those of butyl rubber-based compounds filled with carbon black. However, for this study, it was important to examine unfilled mixtures to specifically evaluate the effect of the modification, namely the influence of the introduction of halogen on the mechanical properties of the rubber. As the chlorine content increased, the mechanical properties of the CIIR vulcanizates, such as tensile strength and elongation at break, also increased. The tensile strength reached its highest value of 1.25 MPa when the amount of chlorine was 15 wt.%. This was 37% higher than that of pure IIR (0.91 MPa). The hardness increased with the increase in chlorine content. This can be attributed to the increased cross-link density and degree of unsaturation of the CIIR. The indicators of rebound and hardness, on average, remain at the level of the as-received butyl rubber, and the fluctuations in values are within the range of standard deviation.

## 5. Conclusions

Currently, the only method used in the industry to produce halobutyl rubbers involves the halogenation of IIR using molecular chlorine or bromine in a solution. The analysis of existing production facilities showed that the industrial methods of obtaining halogenated IIR are technologically complex. The method of butyl rubber modification discussed in this study is based on a mechanochemical approach to initiate the modification process induced by solvent swelling in a polychlorinated n-alkane solution. During the modification, samples were obtained with chlorine content ranging from 3 to 15%. After acetone extraction, the halogen content was quantitatively determined with the oxygen flask combustion method and X-ray photoelectron spectroscopy. It was shown that for samples with total chlorine content of up to 6%, there was almost no leaching of chlorine from the samples. The FT-IR spectra exhibit absorption bands at 500–800 cm^−1^ due to the C–Cl bond in the chlorinated units of macromolecules. The ^1^H NMR spectra display signals (3.4, 3.7, 4.1, and 4.2 ppm) due to the protons of carbon atoms bound to the chlorine atom. Both spectral methods confirm that after acetone extraction, the samples contain chemically bound chlorine, and the chemical structure of chlorinated butyl rubbers corresponds to the chemical structure of CIIR of commercial grade. The emergence of an additional signal cluster might be linked to the presence of the unreacted and unextracted chlorine modifier in the rubbers. An analysis of the isothermal oxidation of rubber in the temperature range 150–180 °C showed that the oxygen absorption rate of chlorinated IIR decreased by 40% compared to the as-received IIR. Halogenated butyl rubber with fewer active double bonds due to chlorination shows better thermal and oxidative resistance, whereas butyl rubber or any other unsaturated polymer with electron-donating groups attached to the carbon atom adjacent to the double bonds is vulnerable to chain scission. The degree of swelling decreases with the increase in the chlorine content. The swelling resistance of rubber depends on the extent of the elastomer’s polarity. The more polar groups attached to the rubber molecules, the higher the polarity and the better the swelling resistance. As the chlorine content increased, the tensile strength of the non-reinforced CIIR vulcanizates also increased. Thus, it was demonstrated that polychlorinated n-alkane, specifically chlorinated paraffins, can be effectively used for the halogenation of butyl rubbers using the mechanochemical modification method induced by solvent swelling. This technology allows the production of a chlorinated rubber solution that can be directly used by rubber goods manufacturers and suppliers.

## Figures and Tables

**Figure 1 polymers-15-04137-f001:**
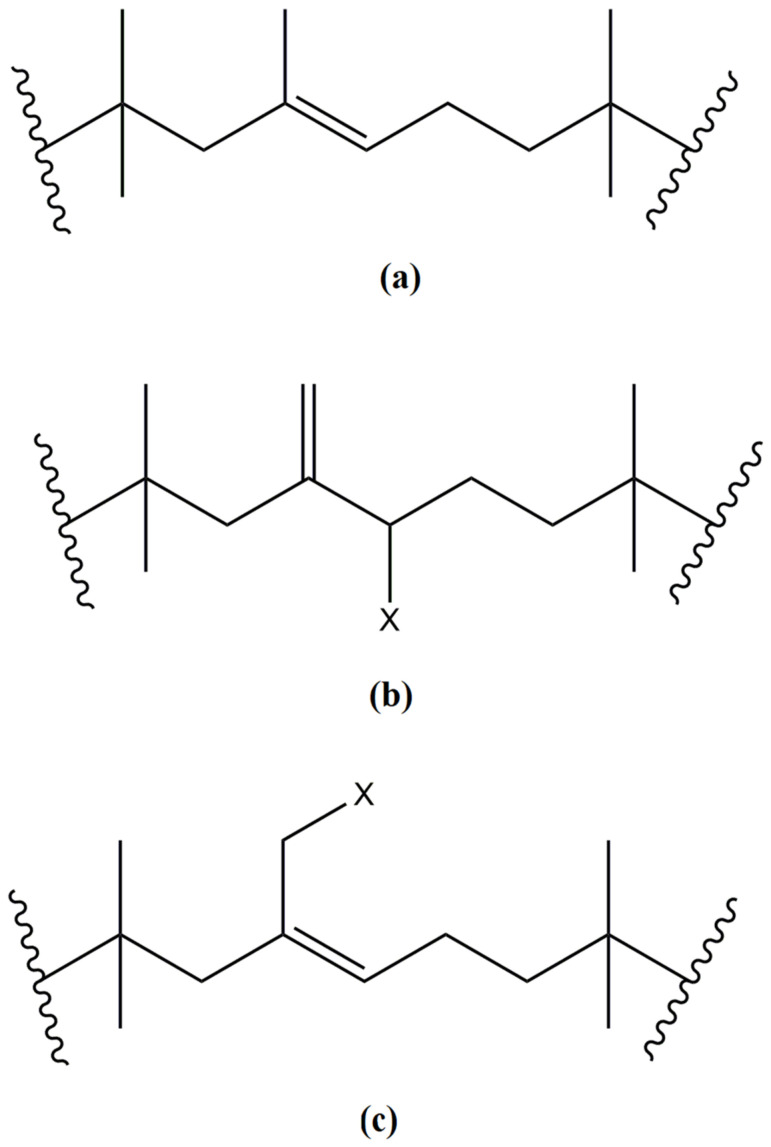
Structures of butyl rubber (IIR) (**a**), exo-methylene halobutyl rubber (EXO), (**b**) and endo-halomethyl butyl rubber (ENDO); X = Cl, Br, or I), (**c**).

**Figure 2 polymers-15-04137-f002:**
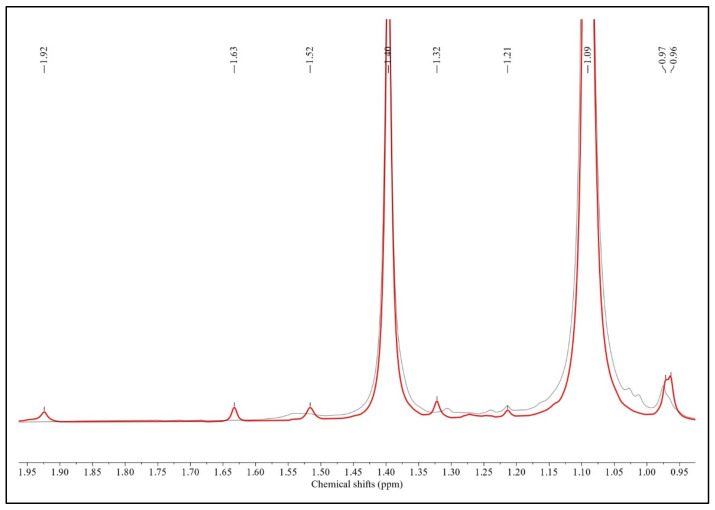
^1^H NMR spectra of IIR (red line) and CIIR-15 (black line; paraffinic zone).

**Figure 3 polymers-15-04137-f003:**
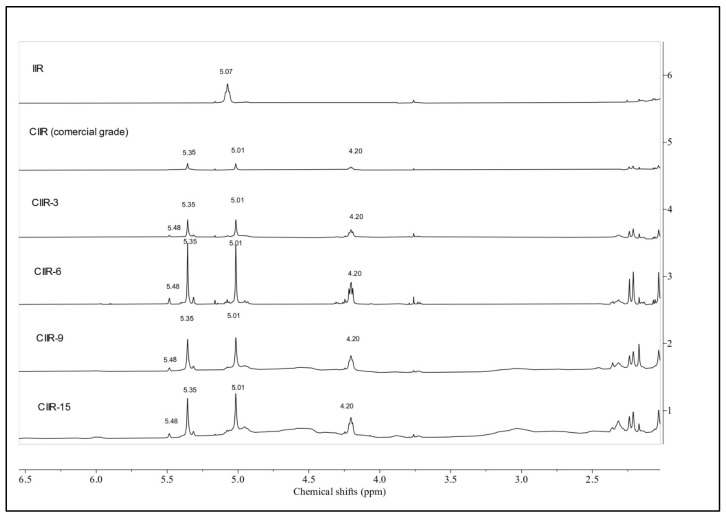
^1^H NMR spectra of IIR, CIIR-3, CIIR-6, CIIR-9, and CIIR-15 (olefinic zone).

**Figure 4 polymers-15-04137-f004:**
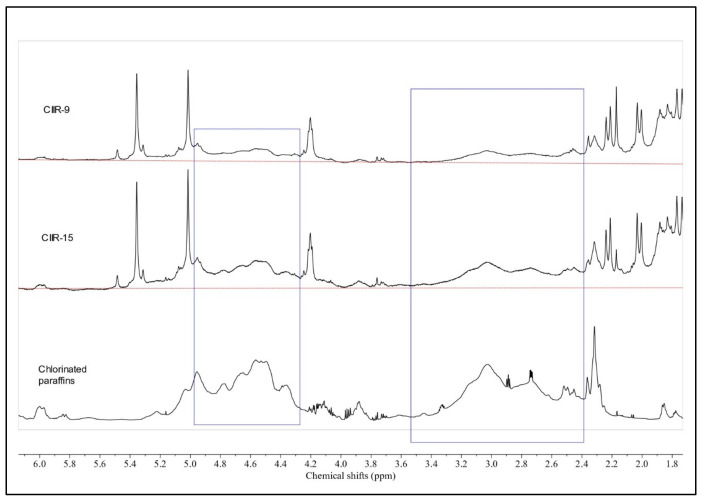
^1^H NMR spectra of CIIR-9, CIIR-15, and chlorinated paraffin.

**Figure 5 polymers-15-04137-f005:**
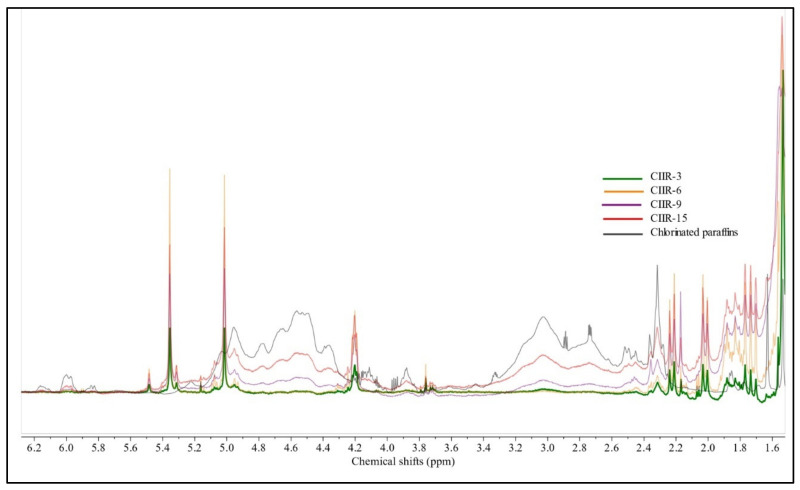
Superimposed ^1^H NMR spectra of CIIR-3, CIIR-6, CIIR-9, CIIR-15, and chlorinated paraffin.

**Figure 6 polymers-15-04137-f006:**
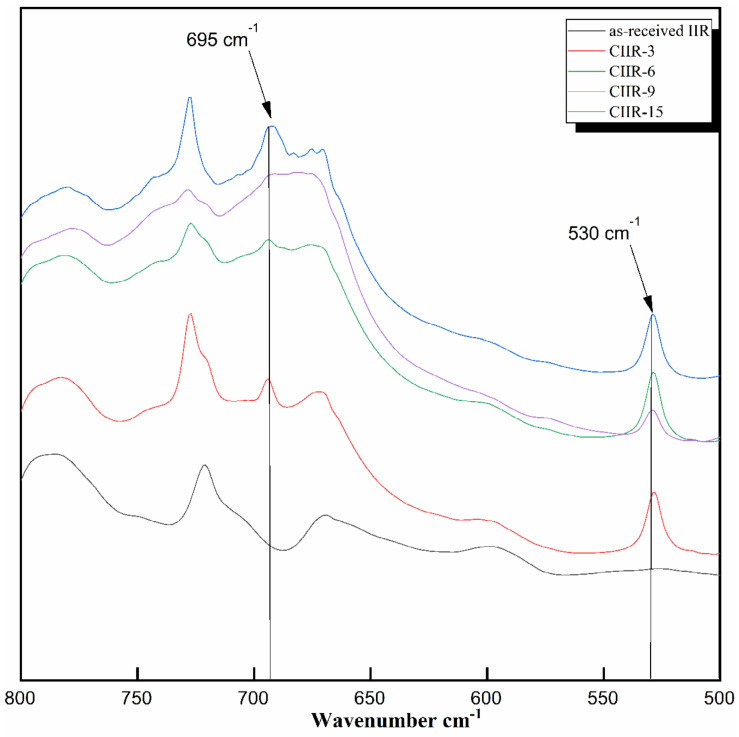
FT-IR spectra of as-received IIR and chlorinated IIR (CIIR-3, CIIR-6, CIIR-9, and CIIR-15) in the wavelength range 500–800 cm^−1^.

**Figure 7 polymers-15-04137-f007:**
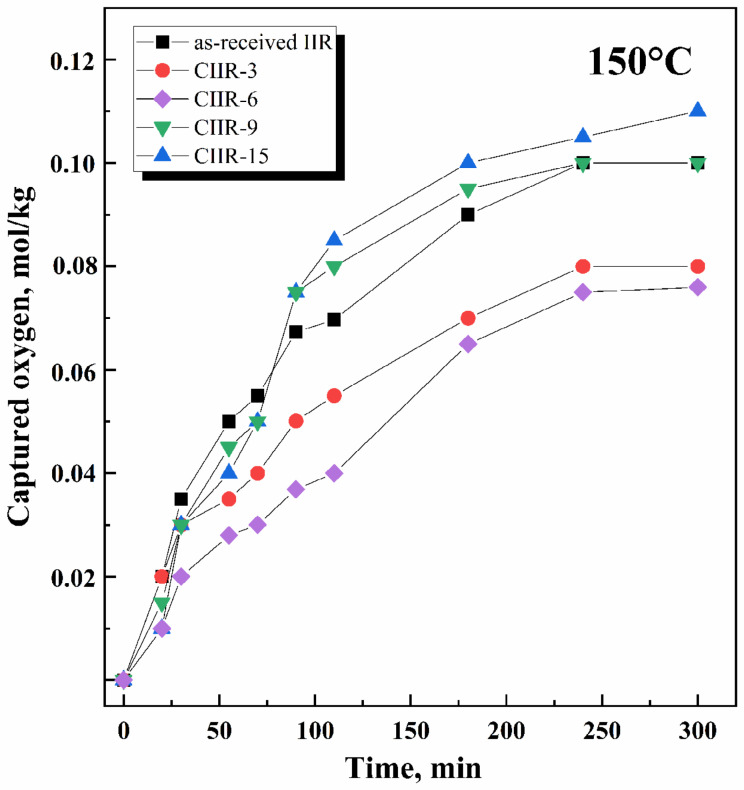
Oxygen absorption kinetics for as-received IIR and chlorinated IIR with different chlorine amounts (CIIR-3, CIIR-6, CIIR-9, and CIIR-15) at temperature 150 °C and oxygen pressure 300 Torr.

**Figure 8 polymers-15-04137-f008:**
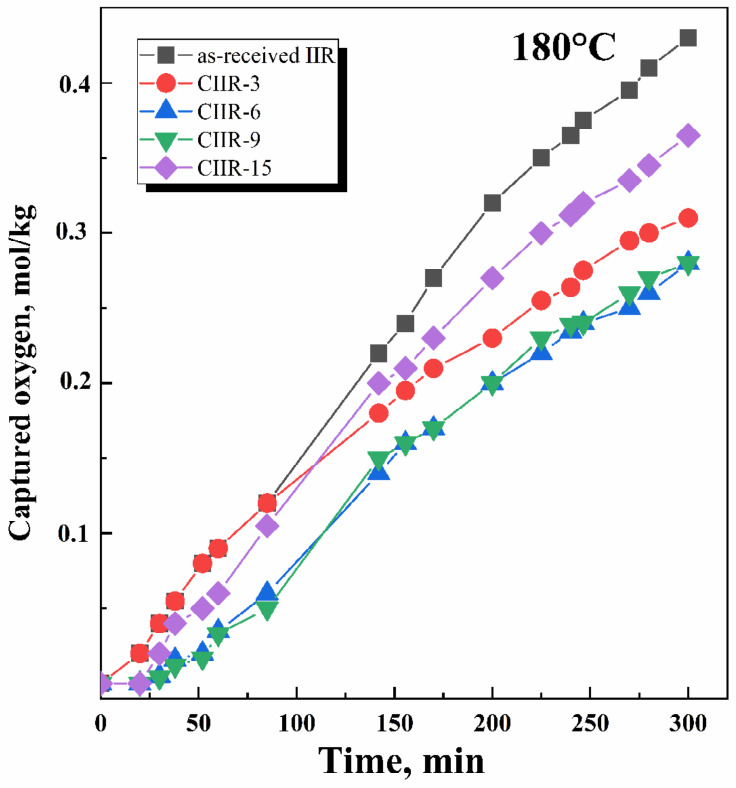
Oxygen absorption kinetic curves for as-received IIR and chlorinated IIR with different chlorine amounts (CIIR-3, CIIR-6, CIIR-9, and CIIR-15) at temperature 180 °C and oxygen pressure 300 Torr.

**Figure 9 polymers-15-04137-f009:**
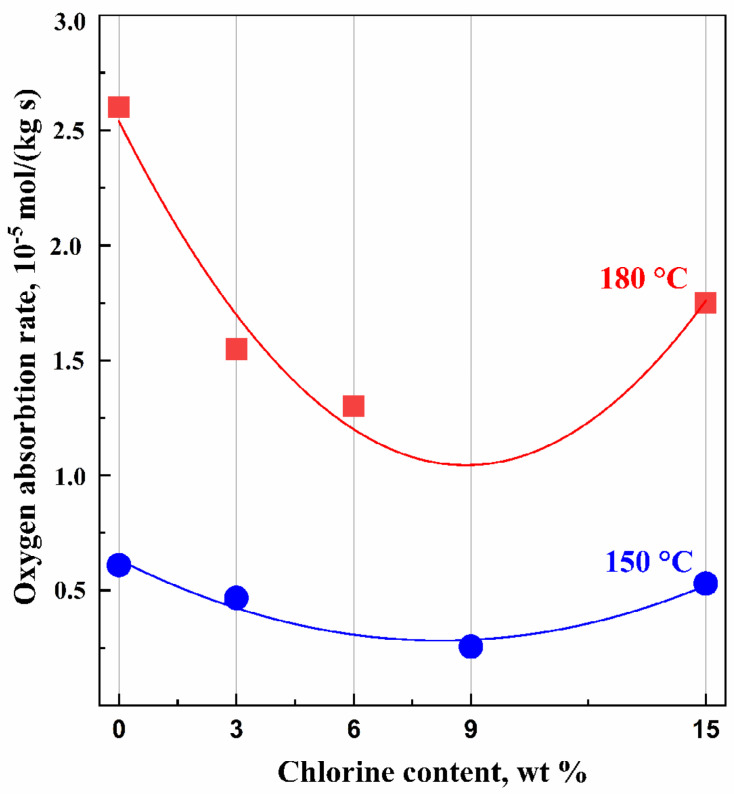
The dependence of the oxidation rate on the chlorine content at temperatures 150 °C and 180 °C and oxygen pressure 300 Torr.

**Figure 10 polymers-15-04137-f010:**
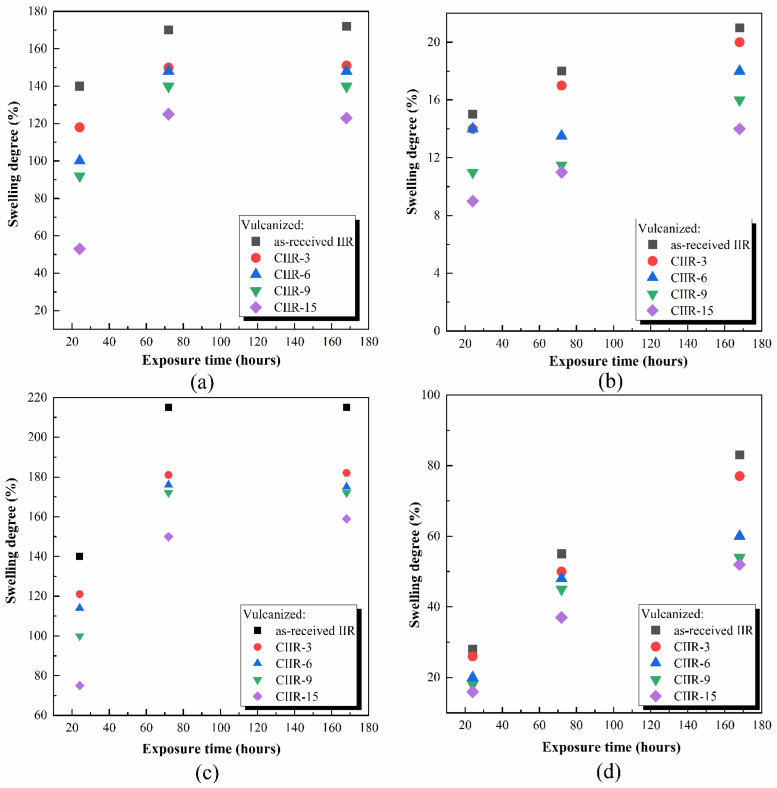
Swelling versus time for as-received IIR and CIIR-3, CIIR-6, CIIR-9, and CIIR-15 vulcanizates in (**a**)—petroleum solvent, (**b**)—nitric acid, (**c**)—toluene, and (**d**)—engine mineral oil.

**Table 1 polymers-15-04137-t001:** Properties of IIR and CIIR rubber.

Parameter	IIR	CIIR
Mooney viscosity	45–56	34–44
Non-saturation level, mol%	1.4–1.8	-
Tg, °C	−69	−42
Chlorine content, % wt.	-	1.1–1.3

**Table 2 polymers-15-04137-t002:** Modification formulation.

Sample	IIR (g)	CP (g)	Calculated Cl Content (*w*/*w*%)
CIIR-3	10	0.209	3 *w*/*w*%
CIIR-6	10	0.417	6 *w*/*w*%
CIIR-9	10	0.626	9 *w*/*w*%
CIIR-15	10	1.043	15 *w*/*w*%

**Table 3 polymers-15-04137-t003:** Modification formulation.

Materials	Compound (phr) *
Rubber	100
Quinol ester of p-quinone dioxime	6
Dithiophosphate accelerator	2

* phr—parts per hundred rubber.

**Table 4 polymers-15-04137-t004:** The total chlorine content [weight (%)] before and after extraction.

Sample	Total Chlorine Content (Weight (%))	
	Before Extraction	After Extraction
CIIR-3	3.0	2.8
CIIR-6	6.0	5.7
CIIR-9	9.0	8.3
CIIR-15	15.0	14.6

**Table 5 polymers-15-04137-t005:** The energy-dispersive X-ray analysis of CIIR-6, CIIR-9, and CIIR-15 before and after extraction.

Sample	Element	Before Extraction			After Extraction		
		Atoms (%)	Weight (%)	Error (±)	Atoms (%)	Weight (%)	Error (±)
CIIR-6	Cl	5.93	14.98	0.50	3.23	8.97	0.46
	C	94.37	85.02	0.74	96.77	91.03	0.90
CIIR-9	Cl	9.01	22.62	0.77	5.28	14.12	0.80
	C	90.99	77.38	0.83	94.72	85.88	1.11
CIIR-15	Cl	12.14	29.98	0.92	7.70	19.77	0.66
	C	87.86	71.02	0.80	92.3	80.23	0.79

**Table 6 polymers-15-04137-t006:** The change in functional groups of IIR, CIIR (commercial grade), CIIR-3, CIIR-6, CIIR-9, and CIIR-15.

Assignment		Chemical Shift, ppm	Multiplicity ^a^	IIR	CIIR (c.g)	CIIR-3	CIIR-6	CIIR-9	CIIR-15
CDCl_3_		δ 7.26	s	v.st.	v.st	v.st	v.st	v.st	v.st
Residual signal (chlorinated paraffins)		δ 6.8	d	N/A	N/A	v.w.	v.w.	v.w.	v.w.
		δ 5.99	d	N/A	N/A	v.w.	v.w.	v.w.	v.w.
-CH_2_-C(=CH_2_)-CHCl-CH_2_		δ 5.36	d	N/A	w	w	st	st	st
Olefinic protons of isoprene units in the 1,4 linkage =CH of isoprene -CH_2_-C(CH_3_)=CH-CH_2_-		δ 5.07 (δ 5.05)	t	st	N/A (o/l)	N/A (o/l)	N/A (o/l)	N/A (o/l)	N/A (o/l)
-CH_2_-C(=CH_2_)-CHCl-CH_2_-		δ 5.01	s	N/A	w	st	st	st	st
Olefinic protons of “branched” isoprene units		δ 4.93 δ 4.98	d	w	v.w.	w	w	w	w
-CH_2_-C(=CH_2_)-CHCl-CH_2_-		δ 4.20	t	N/A	v.w.	w	st	st	st
CH_2_ of isoprene		δ 1.94	s	st	N/A	N/A	N/A	N/A	N/A
CH_3_ of isoprene		δ 1.65	s	st	N/A	N/A	N/A	N/A	N/A
Methyl and methylene protons of the polyisobutylene units of butyl rubber	CH_2_ of isobutylene	δ 1.41	s	v.st	v.st	v.st	v.st	v.st	v.st
	CH_3_ of isobutylene	δ 1.11	s	v.st	v.st	v.st	v.st	v.st	v.st

^a^ s = singlet; d = doublet; t = triplet; N/A—not applicable; N/A (o/l)—not applicable, overlapped; v.st.—very strong; st—strong; w—weak; v.w.—very weak.

**Table 7 polymers-15-04137-t007:** Normalized integral numbers (relative number of protons) of CIIR (commercial grade) and CIIR-3, CIIR-6, CIIR-9, and CIIR-15.

Sample	δ 1.5–0.67	δ 4.20 (4.23–4.16)	δ 5.01 (5.05–4.98)	δ 5.36 (5.4–5.29)	δ 5.49 (5.5–5.46)
CIIR (commercial grade)	100	0.14	0.13	0.17	0.001
CIIR-3	100	0.13	0.16	0.17	0.01
CIIR-6	100	0.15	0.17	0.18	0.02
CIIR-9	100	0.14	0.19	0.18	0.02
CIIR-15	100	0.18	0.26	0.16	0.02

The value of the integrated area in the δ 4.2–δ 4.16 and δ 5.05–δ 4.98 ppm regions indicates a relative increase in the concentration of the halogenated units in the structures of samples with the introduced CP after their extraction in the Soxhlet apparatus. As the amount of added paraffin increases, the normalized integral numbers in the considered areas also increase, especially when compared with the comparison sample (CIIR, commercial grade).

**Table 8 polymers-15-04137-t008:** Mechanical properties of IIR and CIIR vulcanizates with different amounts of chlorine (unfilled).

Samples	Tensile Strength (MPa)	SD * (MPa)	Elongation at Break (%)	SD * (%)	Rebound, (%)	SD * (%)	Hardness (c.u.)	SD * (c.u.)
IIR	0.91	0.07	370	25	21	1	23	2
CIIR-3	1.08	0.04	300	21	18	2	25	1
CIIR-6	1.07	0.05	300	23	21	1	26	1
CIIR-9	1.04	0.11	350	34	22	1	24	2
CIIR-15	1.25	0.09	360	29	20	2	29	2

* SD—standard deviation.

## Data Availability

Not applicable.

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
