# Peer review of "Solvent Swelling-Induced Halogenation of Butyl Rubber Using Polychlorinated N-Alkanes: Structure and Properties"

_polymers, 2023, doi:10.3390/polym15204137_

Round 1
Reviewer 1 Report
This manuscript describes method for the halogenation of isobutylene–isoprene rubber to enhance its stability and resistance to thermal oxidation and aggressive media. The introduction describes methods of modification of butyl rubber.
In my opinion the publication requires many corrections before it can be published.
Authors should indicate the purpose of their research more precisely, especially in Abstract and Section 1. In addition, the abstract does not present the idea of the publication well. It is not very specific and requires precise development to clearly outline the purpose of the research, the scope of the research and the results obtained.
In addition, I have several questions and comments:
1. Why did the authors use this particular method of modifying (solvent swelling-induced halogenation) butyl rubber?
2. Why was chlorinated paraffin (CP) chosen for testing?
3. Tables 1 and 2 may be combined. In Table 2, please add Tg of CIIR.
4. Lines 164-168: The authors write that a magnetic stirrer was used for 3 days during the IIR modification. Actually? What was the cost of such a modification?
5. How was the efficiency of the described IIR modification? What are the chances of using the described method in industrial conditions? Is it even profitable?
6. Why was p-quinone dioxime used as a cross-linking agent to prepare IIR compounds? Why was it used at 6 phr? Is this a safe substance?
7. Why was a dithiophosphate accelerator used to prepare IIR compounds?
8. Lines 216-219: Authors should provide exact characteristics of the ingredients used (including their manufacturers).
9. Please use one form: butyl rubber or isobutylene-isoprene rubber.
10. Was 1H NMR spectroscopy method performed on samples after extraction with acetone? Only extracted samples should be tested.
11. Was FT-IR spectroscopy performed on samples after extraction with acetone? Only extracted samples should be tested.
12. Lines 412-426: The authors described the resistance of the manufactured materials to aggressive media. What is the point of testing such resistance? It is obvious that the incorporation of polar groups into rubber increases its resistance to non-polar solvents.
13. Table 6: The results shown in this table are poor. What is the point of presenting them? What do they contribute to the development of science? Research on compositions containing such modified rubbers and active fillers will make more sense and be more important. It is worth considering supplementing the tests of filled vulcanizates.
14. Table 6: Each result must be reported with standard deviation.
15. Line 436: The authors write about cross-linking degree. How was cross-linking degree tested?
16. Lines 438-440: The hardness conclusions are incorrect. All hardness results are comparable, they only differ within the margin of error.
17. Please rework your conclusions, taking into account the improvement of the article.
18. Please correct the references according to the journal's guidelines.
19. Only 5 references are from the last five years. It's only 16%. Please supplement your literature review with the latest literature.
For the reasons explained before, my recommendation for this paper is accept with major revision. It is noted that the present manuscript needs careful editing.
Author Response
|
Response to Reviewer 1 Comments
|
||
|
1. Summary |
|
|
|
Thank you very much for taking the time to review this manuscript. Please find the detailed responses below and the corresponding corrections and highlighted changes in the re-submitted files. |
||
|
2. Questions for General Evaluation |
Reviewer’s Evaluation |
Response and Revisions |
|
Does the introduction provide sufficient background and include all relevant references? |
Must be improved |
|
|
Are all the cited references relevant to the research? |
Can be improved |
|
|
Is the research design appropriate? |
Must be improved |
|
|
Are the methods adequately described? |
Can be improved |
|
|
Are the results clearly presented? |
Must be improved |
|
|
Are the conclusions supported by the results?
|
Must be improved
|
|
|
3. Point-by-point response to Comments and Suggestions for Authors |
||
|
Comments 1: Why did the authors use this method of modifying (solvent swelling-induced halogenation) butyl rubber? |
||
|
Response 1: Thank you for your comments. The method of halogenation of butyl rubber examined in this work represents the development of the approach to mechanochemical solid-phase halogenation, laid out in the works of Andriasyan et al. (References to the works are added to the article: ref. 17,18,32,33). The principle was based on solid-phase mechanochemical halide modification, initiated by shear stresses during rubber processing in a rubber mixer in the presence of chlorinated paraffins. The method of mechanochemical solid-phase halide modification, initiated by shear stresses, has significantly improved the technology of halogenating elastomers by reducing the number of stages, speeding up the process, and increasing the level of technological and environmental safety of production. However, it has the following disadvantages: · Efficient mixing of the viscous elastomer mass requires high energy consumption; · Difficulty in obtaining a homogeneous product; · The process is accompanied by the self-heating of the reaction mass, therefore strict process control is required, and, if necessary, measures to cool the mixing equipment; · This method is limited in terms of the halogen content in the modified rubbers. Based on the accumulated literature data on the consideration of rubber swelling as a mechanochemical process during which the polymer experiences high stress, the possibility of conducting mechanochemical halide modification under the influence of swelling pressure was studied. Moreover, the technology holds practical significance. Often, chlorinated butyl rubbers are needed in solution form for uses like adhesives and protective rubber coatings. Therefore, it was fitting to develop a modification method that produces halogenated rubber in solution form. The added information can be found on lines 116-136; 172-182.
|
||
|
Comments 2: Why was chlorinated paraffin (CP) chosen for testing? |
||
|
Response 2: Chlorinated paraffins are widely available raw materials, a modifier containing a high amount of chlorine (70%). Chlorinated paraffins are classified as low-hazard substances (hazard class 4) and do not require additional protective measures during handling (unlike gaseous chlorinating compounds). They are also relatively inexpensive raw materials. Previously, the effectiveness of using this compound for halogenation purposes was demonstrated (Ref.: 32,33). The added information can be found on lines 202-205.
Comments 3: Tables 1 and 2 may be combined. In Table 2, please add Tg of CIIR. Response 3: Agree. This change was made to the text (tables were merged, and data about Tg was added).
Comments 4: Lines 164-168: The authors write that a magnetic stirrer was used for 3 days during the IIR modification. Actually? What was the cost of such a modification? Response 4: The concentration of the butyl rubber solution in toluene was 15 wt.%. Butyl rubber solutions have high viscosity, and the dissolution process goes through the swelling stages of butyl rubber (pieces of rubber, size 1 cm2.). It can be confidently stated that a homogeneous rubber solution is achieved after 3 days. However, this time can be reduced by using more powerful overhead Stirrers and finer rubber cuts.
Comments 5: How was the efficiency of the described IIR modification? What are the chances of using the described method in industrial conditions? Is it even profitable? Response 5: This technology allows to produce a chlorinated rubber solution that can be directly used for rubber goods manufacturers and suppliers to produce adhesive bonds, rubber coatings or for spreading on fabrics when creating rubber pneumatic structures. The advantage is that one can obtain chlorinated butyl rubber based on readily available butyl rubber and an accessible chlorine-containing modifier, replacing industrial chlorinated butyl rubber (CIIR) in production. Thus, rubber goods manufacturers can partially replace industrial CIIR with modified butyl rubber in products where a solution of chlorinated butyl rubber is required, without compromising the quality of the product. Information on the possibility of applying this technology to production has been added to the Introduction section (lines 172-182).
Comments 6: Why was p-quinone dioxime used as a cross-linking agent to prepare IIR compounds? Why was it used at 6 phr? Is this a safe substance? Response 6: Quinoline esters including p-quinone dioxime are effectively used for the vulcanization of non-conjugated rubbers, including butyl rubber. The use of p-quinone dioxime as adhesion promoters in chlorinated rubber-based adhesives for bonding rubber to metal is also known. The use of quinoid vulcanizing systems allows to produce rubbers resistant to the action of liquid aggressive media. Using quinoline esters as vulcanizing agents leads to increased resistance of samples to the effects of nitric and sulfuric acids. P-quinone dioxime belongs to the green curing agent (Cent. Eur. J. Energ. Mater. 2022, 19(1): 18-38; DOI 10.22211/cejem/147553) The use of this vulcanizing agent was explained by its ability to fully dissolve in aromatic solvents. An explanation for the choice of this vulcanizing agent has been added to the section "Characterization of this vulcanizing agent" (lines 270-278) The work to determine the optimal concentration of the vulcanizing agent and vulcanization accelerator was conducted earlier, and it was established that 6 phr of the p-quinone dioxime and 2 phr of the dithiophosphate accelerator correspond to the composition of the optimal vulcanizing group for the solution of chlorinated butyl rubber. A reference to the work has been added to the article text (ref. 37).
Comments 7: Why was a dithiophosphate accelerator used to prepare IIR compounds? Response 7: The dithiophosphate accelerator is a liquid accelerator, which dissolves excellently in a butyl rubber solution.
Comments 8: Lines 216-219: Authors should provide exact characteristics of the ingredients used (including their manufacturers). Response 8: Characteristics of the ingredients used have been added to the text, as well as information about the manufacturers ((lines 270-278)
Comments 9: Please use one form: butyl rubber or isobutylene-isoprene rubber. Response 9: Throughout the article, "isobutylene-isoprene rubber" was replaced with "butyl rubber".
Comments 10: Was 1H NMR spectroscopy method performed on samples after extraction with acetone? Only extracted samples should be tested. Response 10: Yes, 1H NMR spectroscopy method was performed only on extracted samples.
Comments 11: Was FT-IR spectroscopy performed on samples after extraction with acetone? Only extracted samples should be tested. Response 11: Yes, FT-IR spectroscopy method was performed only on extracted samples.
Comments 12: Lines 412-426: The authors described the resistance of the manufactured materials to aggressive media. What is the point of testing such resistance? It is obvious that the incorporation of polar groups into rubber increases its resistance to non-polar solvents. Response 12: It was important to demonstrate the practical effectiveness of such a modification method for altering the properties of the resulting materials. The modification not only changes the structure of the rubber but also significantly affects its properties.
Comments 13: Table 6: The results shown in this table are poor. What is the point of presenting them? What do they contribute to the development of science? Research on compositions containing such modified rubbers and active fillers will make more sense and be more important. It is worth considering supplementing the tests of filled vulcanizates. Response 13: Indeed, the mechanical characteristics of these mixtures are low compared to butyl rubber-based compounds filled with carbon black. However, for this study, it was important to examine non-filled mixtures specifically to assess the impact of the modification (the introduction of halogen on the mechanical properties of the rubber). In filled mixtures, there was a likelihood of not observing the halogen influence trend as distinctly. This explanation was added to the article's text to make the motivation for using non-filled vulcanizates clear to the reader (lines 498-501).
Comments 14: Table 6: Each result must be reported with standard deviation. Response 14: A column with standard deviation has been added to the table.
Comments 15: Line 436: The authors write about cross-linking degree. How was cross-linking degree tested? Response 15: Previously, the rheological characteristics of the mixtures were studied using the Monsanto rheometer (Ref.37). Based on the relationship between the minimum and maximum torque, one can indirectly infer the increase in the degree of rubber crosslinking with the increase in the amount of modifier.
Comments 16: Lines 438-440: The hardness conclusions are incorrect. All hardness results are comparable, they only differ within the margin of error. Response 16: Changes have been made to the description of mechanical properties results.
Comments 17: Please rework your conclusions, taking into account the improvement of the article. Response 17: The conclusion was reworked in accordance with the recommendations.
Comments 18: Please correct the references according to the journal's guidelines. Response 18: The references was corrected according to the journal's guidelines.
Comments 19: Only 5 references are from the last five years. It's only 16%. Please supplement your literature review with the latest literature. Response 19: Recent publications have been added to the References.
|
||

Reviewer 2 Report
The reviewer wrote his comments in the attached file. Please find them.

Author Response
|
Response to Reviewer 2 Comments
|
||
|
1. Summary |
|
|
|
Thank you very much for taking the time to review this manuscript. Please find the detailed responses below and the corresponding corrections and highlighted changes in the re-submitted files. |
||
|
2. Questions for General Evaluation |
Reviewer’s Evaluation |
Response and Revisions |
|
Does the introduction provide sufficient background and include all relevant references? |
Yes |
|
|
Are all the cited references relevant to the research? |
Yes |
|
|
Is the research design appropriate? |
Can be improved |
|
|
Are the methods adequately described? |
Can be improved |
|
|
Are the results clearly presented? |
Yes |
|
|
Are the conclusions supported by the results?
|
Yes
|
|
|
3. Point-by-point response to Comments and Suggestions for Authors |
||
Comments 1: Chemical structure of IIR and XIIR should be illustrated.
Response 1: Thank you very much for this comment. We have added images of the chemical structures of IIR and CIIR to the Introduction section (line 189-191)
Comments 2: Line 135 (like CCl4)
Response 2: The text has been corrected, and the changes have been made (line 167)
Comments 3: If a specific chlorinated solution (such as CCl4, C2Cl6 not a mixture (CP) was used, are there any difference of CIIR obtained?
Response 3: In this work, only CP was used for halogenation purposes. However, in another study of ours, the possibility of using 1,1,2‐trifluoro‐1,2,2‐trichlor‐ethane for halide modification of the surface of butadiene-nitrile rubbers was demonstrated. Rubbers based on butadiene-nitrile rubber have a high capacity for limited swelling in this compound (10.1002/pen.26413)
Comments 4: Toluene solution of CP was used. When liquid neat chlorinated alkane such as CCl4 was used, what kind of CIIR was obtained?
Response 4: Thank you for the interesting question. Within the scope of this work, this issue was not studied. The possibility of modifying other solvents capable of swelling in chlorine-containing solutions requires further research.
Comments 5: How to control the chlorine contents?
Response 5: The change in the amount of chlorine is regulated by adjusting the modifier dosage, based on the assumption that 1 gram of the modifier contains 0.695 grams of chlorine, and the mass concentration of chlorine is calculated per 100 parts by weight of rubber.
Comments 6: What is Kc?
Response 6:
The correction coefficient Kс for adjusting the concentration of the Hg(NO3)2 solution precisely to 0.02 N is calculated using the formula:
where:
m – mass of NaCl, mg;
V – volume of the 0.02 N solution of Hg(NO3)2 used for titration, ml;
1.1690 – mass of NaCl corresponding to 1 ml of 0.02 N solution of Hg(NO3)2, mg.
Clarifying information was added to the "Methods" section (line 200-206).
Comments 7: Why so different? (-0,3% and -2,70%)
Response 7. It is most likely that this difference may be due to the local heterogeneity of this sample of modified rubber. In the sample analyzed by the Oxygen flask combustion method, the film after extraction might have been slightly thinner, and some of the unreacted modifier might have remained unextracted. However, despite the difference in data for the CIIR-6 sample, both methods, characterized by different degrees of accuracy, show the presence of chlorine after acetone extraction.
Comments 8: Is the HCl detectable?
Response 8: Within the scope of this work, the task of capturing HCl during dehydrochlorination was not posed. In the future, we plan to identify the pyrolysis products of the samples using the method of quantitative flash pyrolysis FT-IR spectroscopy. Based on the experimental observations, it can be confidently stated that samples with 9% and 15% chlorine visually change color significantly during the thermal oxidation experiment (from light to dark yellow)
Corrections related to stylistic formatting have been made in the article at the appropriate places and are highlighted in red.

Round 2
Reviewer 1 Report
I accept this paper
Author Response
Thank you very much for your help in improving the article!
Reviewer 2 Report
The reviewer wrote his comments in the attached file. Please find them.

Author Response
For research article
|
Response to Reviewer 2 Comments
|
||
|
1. Summary |
|
|
|
Thank you very much for taking the time to review this manuscript. Please find the detailed responses below and the corresponding corrections and highlighted changes in the re-submitted files. |
||
|
2. Questions for General Evaluation |
Reviewer’s Evaluation |
Response and Revisions |
|
Does the introduction provide sufficient background and include all relevant references? |
Yes |
|
|
Are all the cited references relevant to the research? |
Yes |
|
|
Is the research design appropriate? |
Yes |
|
|
Are the methods adequately described? |
Can be improved |
|
|
Are the results clearly presented? |
Can be improved |
|
|
Are the conclusions supported by the results?
|
Yes
|
|
|
3. Point-by-point response to Comments and Suggestions for Authors |
||
For convenience, new changes in the text were highlighted in green.
Comments 1: line 193 (IIR) (EXO)
Response 1: Corrections have been made to the text.
Comments 2: line 216. The modifier = CP?
Response 2: The word 'the modifier' has been replaced with 'CP'.
Comments 3: line 217. Amount?
Response 3: The amount of toluene was 70 ml. Clarification has been added to the text.
Comments 4: line 221. performed?
Response 4: Corrections have been made to the text.
Comments 5: lines 249, 253. Kc subscript
Response 5: Corrections have been made to the text.
Comments 6: line 271
Response 6: Corrections have been made to the text.
Comments 7: Lines 280,281,284 (parts per hundred rubber)
Response 7: The text has been corrected. A footnote has been added to the table.
Comments 8: Line 360 (EXO)
Response 8: The text has been corrected. (the exo-methylene unit (EXO))
Comments 9: Lines 367
Response 9: The text has been corrected.
Comments 10: Lines 368-373
Response 10: An explanation regarding the signal at 5.05 ppm has been added to the text.
Comments 11: Line 395. Table 4
Response 11: The table has been formatted.
Comments 12: Line 422. Fig.6
Response 12: The order of sample display in the legend has been changed.
Comments 12: References
Response 12: The formatting of 'References' has been corrected.
Thank you very much for your help in improving the article!